# ToolFlowNet: Robotic Manipulation with Tools via Predicting Tool Flow from Point Clouds

**Daniel Seita, Yufei Wang[†], Sarthak J. Shetty[†], Edward Yao Li[†], Zackory Erickson, David Held**
[†]Equal contribution.
The Robotics Institute, Carnegie Mellon University, USA
Correspondence to: `dseita@andrew.cmu.edu`

**Abstract:** Point clouds are a widely available and canonical data modality which convey the 3D geometry of a scene. Despite significant progress in classification and segmentation from point clouds, policy learning from such a modality remains challenging, and most prior works in imitation learning focus on learning policies from images or state information. In this paper, we propose a novel framework for learning policies from point clouds for robotic manipulation with tools. We use a novel neural network, ToolFlowNet, which predicts dense per-point flow on the tool that the robot controls, and then uses the flow to derive the transformation that the robot should execute. We apply this framework to imitation learning of challenging deformable object manipulation tasks with continuous movement of tools, including scooping and pouring, and demonstrate significantly improved performance over baselines which do not use flow. We perform 50 physical scooping experiments with ToolFlowNet and attain 82% scooping success. See https://tinyurl.com/toolflownet for supplementary material.

**Keywords:** Flow, Point Clouds, Tool Manipulation, Deformables

## 1 Introduction

Recently, learning-based techniques have become effective for improving the generalization capabilities of robots for manipulation tasks such as grasping [1], reorienting [2], rearrangement [3], and tossing [4]. Data observations tend to be either images [5, 6, 7] or state information such as joint angles or end-effector poses [8], which are passed into a deep network to obtain an output vector encoding an action, typically representing a change in end-effector pose or joint angles. While these approaches have shown a wide range of successes, a fundamental limitation has to do with the nature of the observation. Using images requires projecting information into a 2D space which might lose valuable 3D information. Furthermore, learning from images in simulation often leads to a sim2real gap [9] in performance. Although it is easy to access the internal robot states such as joint angles, the robot does not necessarily have the ground-truth state of objects in the environment, which might require complex state estimation systems [10]. Moreover, it is hard to define a state for deformable objects like liquid and cloth [11, 12].

In this work, we propose a framework for learning robotic manipulation from point cloud observations. Point clouds are a canonical data modality and are widely available from camera sensors, providing valuable information about the structure of the 3D space [13, 14]. However, policy learning from point clouds has been less explored compared to learning from images or state, potentially owing to the difficulty of reasoning about raw 3D point cloud inputs. While there have been many proposed architectures which are specialized for learning from point clouds [13, 15, 14, 16, 17], these works tend to focus on computer vision tasks such as classification and segmentation. Policy learning from point clouds, while feasible in some contexts [18, 19], remains challenging.

We study learning from point clouds for robotic manipulation tasks with tools. The input data is a segmented point cloud which, for each point, contains its 3D coordinates and a one-hot vector

6th Conference on Robot Learning (CoRL 2022), Auckland, New Zealand.

| Before | Tool Flow | After | Before | Tool Flow | After |

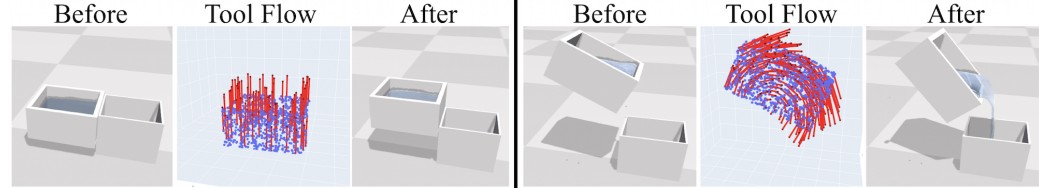

Figure 1: ToolFlowNet applied on a pouring task in simulation, where the tool is the box which contains water. Given a point cloud (colored blue), ToolFlowNet learns dense per-point flow vectors (colored red), which describe the intended 3D motion of each tool point. These are converted to translation and rotation actions. Left: the tool moves upwards. Right: the tool rotates to pour water. We subsample the flow for visual clarity.

indicating the object class the point belongs to. Our key insight is to use *dense representations* and *flow* to represent the tool action. We build upon dense point-cloud processing architectures [15] and train the model to predict per-point output values which we call *tool flow*. This represents the 3D movement of each tool point in a point cloud from one time step to the next, which is an instance of scene flow [20]. Our model is trained with Behavioral Cloning on tool flow data, which provides a dense per-point supervision. Given the set of tool flow vectors, we convert flow to an SE(3) transformation, which represents the actual action a robot would execute. We call this model ToolFlowNet and visualize it in Figure 1 for a pouring task in simulation. We compare this against non-dense methods which directly regress to an action and demonstrate the benefits of tool flow as an action representation. To summarize:

- We propose a general framework for learning from segmented point clouds for manipulation with tools by utilizing a novel architecture, ToolFlowNet, which predicts per-point tool flow vectors.
- We show how to train ToolFlowNet for imitation learning and explore different loss functions for training. We perform extensive ablation studies to validate these choices.
- We perform simulated imitation learning experiments on scooping and pouring tasks and show the benefits of using ToolFlowNet over baselines which do not use flow.
- We demonstrate ToolFlowNet achieves 82% success rate on 50 physical scooping trials.

## 2   Related Work

**Point Clouds and Flow**   Researchers have proposed a variety of architectures specialized for learning from point clouds [13, 15, 14, 21, 17, 22, 23]. We aim to explore policy learning for robotic manipulation from point clouds, and the approach we propose is compatible with any architecture producing per-point outputs from point clouds. Optical flow [24, 25] and its 3D counterpart, scene flow [20, 26], are widely used in computer vision, particularly in autonomous driving setups where the objective is to associate the movement of each pixel (or a point in 3D space) from one image (or point cloud) to the next time step. We use flow as an action representation for robot manipulation, and our method could integrate prior flow estimation techniques if necessary.

**Policy Learning from Point Clouds or Flow for Robotic Manipulation**   Learning from point clouds has been explored in grasping [19, 27], in-hand manipulation (by voxelizing) [18], visual navigation [28], and shaping 3D deformables [29]. Our work differs in that we study tasks that involve manipulating a tool in 3D space from point cloud observations, and where we use tool flow as the action representation to improve learning. While Qin et al. [30] extract tool point clouds and learn keypoints for grasping and manipulating tools, we instead predict dense tool flow for manipulating the tool. Some prior work has explored policy learning using *flow* for robot manipulation, such as for fabric folding [31], manipulating articulated objects [32, 33], and manipulating 3D deformables [34]. This work differs in that we propose a more general framework that does not assume a specific structure of the objects being manipulated, and which predicts flow on the tool a robot controls instead of flow on a target object. Furthermore, unlike prior work [35] which iteratively minimizes flow with pick and place actions, or other work [36] which uses optical flow on tactile sensors, we use flow to derive continuous tool motions in 3D space from visual input. A recent work [37] estimates optical flow using RGBD images from the current frame to the demon-

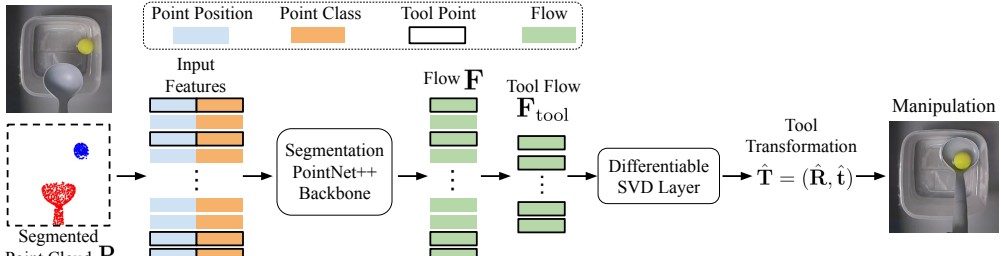

Figure 2: The proposed ToolFlowNet framework learns from segmented point clouds, which form the input to a dense point cloud network to produce per-point flow vectors. We extract just the tool points (bolded above for clarity) and use those tool points to determine the transformation that the robot should apply to the tool. See Section 3 for further details. Above, we show the physical scooping task; see Section 4.4 for details.

stration and extracts a transformation to align them. In contrast, we do not use flow for aligning frames to demonstrations but for deriving the transformations the tool should follow.

**Deformable Object Manipulation**   We apply our proposed tool flow framework on tasks with continuous control of a tool for deformable object manipulation. Such manipulation is challenging for robots for reasons such as the difficulty in specifying a concise state representation for deformables and their complex dynamics [11, 12]. We test on scooping and pouring. Variants of these tasks have been studied in prior work. For example, [38] use scooping as an example application for task and motion planning, and [39] test scooping of granular media using a 2D image representation. Unlike these works, our approach is a general framework for robots performing continuous control of a tool to manipulate deformables in 3D space. Prior works [40, 41, 42, 43, 44] propose methods to detect or model physics properties of granular media or liquids and test on scooping and pouring. In contrast, we propose a general method of predicting 3D tool flow which does not require modeling properties of deformables and is not specialized to scooping or pouring tasks, and which uses point clouds as inputs instead of RGB images [45].

## 3   Method: ToolFlowNet for Behavioral Cloning from Point Clouds

We consider policy learning from segmented point cloud observations. A segmented point cloud $\mathbf{P}_t$ at time $t$ is an $N \times d$ array with $N$ points, each with feature dimension $d$. The feature of the $i$th point $p^{(i)} \in \mathbf{P}_t$ consists of its 3D coordinate position and a one-hot vector indicating the class of the object to which $p^{(i)}$ belongs. For ease of notation, we suppress the time subscript $t$ and the point index superscript $i$ when the distinction is not needed. We study Behavioral Cloning (BC) [46] from segmented point clouds. BC assumes access to a dataset $\mathcal{D} = \{(\mathbf{o}_t, \mathbf{a}_t^*)\}_{t=1}^M$ of observation-action pairs $(\mathbf{o}_t, \mathbf{a}_t^*)$ from a demonstrator, where $\mathbf{o}_t$ indicates any type of observation (of which segmented point clouds are one example). BC performs supervised learning to learn a policy $\pi$ with parameters $\theta$ such that the predicted action $\hat{\mathbf{a}}_t = \pi_\theta(\mathbf{o}_t)$ is close to the ground truth label $\mathbf{a}_t^*$. While prone to compounding errors at test time [47], BC has shown surprising effectiveness when compared to more complex learning-based algorithms in some robotic manipulation contexts [48, 49], which motivates further study on how it can be done with point cloud observations. In future work, we will explore combining our method with other imitation learning algorithms [50, 51, 52].

### 3.1   Tool Flow As Action Representation

We propose to use *tool flow* as an internal representation for the action, where the flow $f^{(i)} \in \mathbb{R}^3$ associated with point $p^{(i)}$ is a 3D vector. For a given tool point, we interpret its flow vector as representing how the point will move in 3D space as a result of "applying" the flow. To form the policy $\pi_\theta$, we use a dense point cloud network (such as a segmentation PointNet++ [15]), which given an input point cloud $\mathbf{P}$ produces per-point outputs. The point cloud input is already segmented in that it contains, for each point, the 3D world position and a one-hot encoding of its class. With an $(N \times d_1)$-sized point cloud $\mathbf{P}$ as input, the output $\mathbf{F}$ has dimension $(N \times d_2)$, where $d_2$ is the output dimension (in our case, $d_2 = 3$). We then extract from the output $\mathbf{F}$ the subset of $N' \leq N$

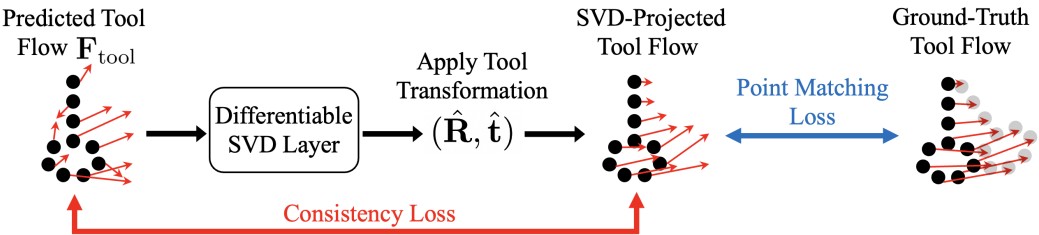

Figure 3: A visualization of the proposed point matching loss (Eq. 3) and consistency loss (Eq. 4). The black points visualize a simplified ladle's point cloud, and the thin red arrows represent the flows on the tool points.

points in $\mathbf{P}$ corresponding to all points on the *tool*, while ignoring points belonging to other object classes. This results in a set of predicted 3D tool flow vectors $\mathbf{F}_{\text{tool}} = \{f^{(i)}\}_{i=1}^{N'}$ with $f^{(i)} \in \mathbb{R}^3$ for each tool point.

Suppose that, at time $t$, the expert applies an action to the tool which is given by a ground-truth transformation $\mathbf{a}^* = (\mathbf{R}^*, \mathbf{t}^*) = \mathbf{T}^* \in SE(3)$. Let $\mathbf{P}_{\text{tool}} \subseteq \mathbf{P}$ be the set of 3D points on the tool. Then the ground-truth tool flow is given by $\mathbf{F}_{\text{gt}} = \mathbf{T}^*\mathbf{P}_{\text{tool}} - \mathbf{P}_{\text{tool}}$ where $\mathbf{T}^*\mathbf{P}_{\text{tool}}$ is the result of applying the transformation $\mathbf{T}^*$ on all points in $\mathbf{P}_{\text{tool}}$. Thus, there is a one-to-one correspondence between the transformation $\mathbf{T}^*$ and flow $\mathbf{F}_{\text{gt}}$; nonetheless, we show in Section 4 that estimating the tool flow leads to improved performance compared to estimating the transformation $\mathbf{T}^*$ directly.

Given the set of predicted 3D tool flow vectors $\mathbf{F}_{\text{tool}}$, the next step is to extract a single overall action $\hat{\mathbf{a}}$, where $\hat{\mathbf{a}}$ is a transformation that represents the change in translation and rotation of the tool's pose. To compute the action, we consider the tool point clouds $\mathbf{P}_{\text{tool}}$ and $\mathbf{P}'_{\text{tool}} = \mathbf{P}_{\text{tool}} + \mathbf{F}_{\text{tool}}$, where in the latter, we move each point based on its estimated flow. Our objective is to estimate the best-fit tool transformation $\hat{\mathbf{T}} = (\hat{\mathbf{R}}, \hat{\mathbf{t}})$ which contains rotation and translation components, respectively, to align $\mathbf{P}_{\text{tool}}$ and $\mathbf{P}'_{\text{tool}}$, i.e., we want to find $\hat{\mathbf{T}}$ to minimize $\|\hat{\mathbf{T}}\mathbf{P}_{\text{tool}} - \mathbf{P}'_{\text{tool}}\|_2$. To obtain the rotation $\hat{\mathbf{R}}$, we first center the two tool point clouds to obtain $\bar{\mathbf{P}}_{\text{tool}}$ and $\bar{\mathbf{P}}'_{\text{tool}}$. We then input the centered point clouds to a differentiable, parameter-less Singular Value Decomposition (SVD) layer [53, 54] which computes the rotation which best aligns $\bar{\mathbf{P}}_{\text{tool}}$ and $\bar{\mathbf{P}}'_{\text{tool}}$ with respect to mean square error (MSE). The change in translation $\hat{\mathbf{t}}$ can then be computed as $\hat{\mathbf{t}} = C(\mathbf{P}'_{\text{tool}}) - \hat{\mathbf{R}}C(\mathbf{P}_{\text{tool}})$, where $C(\mathbf{P})$ denotes the centroid of the point cloud $\mathbf{P}$. By combining the translation and rotation components, we produce the full transformation $\hat{\mathbf{T}}$, which we treat as our action representation for the policy. The outputs for the non-tool points are not supervised. We call the resulting point cloud-to-action system as ToolFlowNet (see Figure 2), which can be used by a robot for manipulation. Mathematically, let $F_\theta$ represent the segmentation PointNet++ that generates the flow vectors. ToolFlowNet computes the tool transformation as follows:

$$\hat{\mathbf{a}} = (\hat{\mathbf{R}}, \hat{\mathbf{t}}) = \pi_\theta(\mathbf{o}) = \text{SVD}(\mathbf{F}_{\text{tool}}) \tag{1}$$

where SVD represents the parameter-less Singular Value Decomposition layer as described above, and $\mathbf{F}_{\text{tool}}$ is the flow corresponding to the tool points in the estimated flow $\mathbf{F} = F_\theta(\mathbf{P})$.

## 3.2 Imitation Learning via Tool Point Matching and Consistency Losses

**Point Matching Loss:** Given the policy's predicted action $\hat{\mathbf{a}} = \pi_\theta(\mathbf{o})$, a straightforward way to imitate the ground truth action $\mathbf{a}^* = (\mathbf{R}^*, \mathbf{t}^*)$ is to use the MSE loss:

$$L_{\text{mse}}(\hat{\mathbf{a}}, \mathbf{a}^*) = \beta_1\|\hat{\mathbf{R}} - \mathbf{R}^*\|_2 + \beta_2\|\hat{\mathbf{t}} - \mathbf{t}^*\|_2, \tag{2}$$

where $\beta_1$ and $\beta_2$ are weights for the translation and rotation components. Instead of trying to balance the weights, in this paper, we use a point matching loss to reduce the discrepancy between $\hat{\mathbf{a}} = (\hat{\mathbf{R}}, \hat{\mathbf{t}})$ and the ground truth action $\mathbf{a}^* = (\mathbf{R}^*, \mathbf{t}^*)$. Given the predicted action, the transformation $(\hat{\mathbf{R}}, \hat{\mathbf{t}})$ is applied on all the original $N'$ tool points in the point cloud, and the loss function $L_{\text{point}}$ computes the Euclidean distance between the tool points transformed using the predicted action

$(\hat{\mathbf{R}}, \hat{\mathbf{t}})$ versus the tool points transformed using the ground-truth action $(\mathbf{R}^*, \mathbf{t}^*)$:

$$L_{\text{point}}(\mathbf{P}, \hat{\mathbf{a}}, \mathbf{a}^*) = \frac{1}{N'} \sum_{i=1}^{N'} \|(\hat{\mathbf{R}}p^{(i)} + \hat{\mathbf{t}}) - (\mathbf{R}^*p^{(i)} + \mathbf{t}^*)\|_2, \tag{3}$$

where $p^{(i)}$ iterates through the $N'$ tool points in $\mathbf{P}_{\text{tool}} \subseteq \mathbf{P}$, and we interpret $\hat{\mathbf{R}}$ and $\mathbf{R}^*$ as representing $3 \times 3$ rotation matrices. Prior work on 6D pose estimation [55, 56, 57] has used variants of this loss function to jointly optimize for translation and rotation as compared to balancing the weights on separate translation and rotation losses. Our usage of $L_{\text{point}}$ is similar to that in Wang et al. [19] where the matching loss is on tool points directly controllable by the robot.

**Consistency Loss:** While $L_{\text{point}}$ should allow the policy $\pi_\theta$ to learn SE(3) pose changes (and thus, actions), its effect on optimizing the predicted flow vectors $\mathbf{F}$ happens via backpropagating through a differentiable SVD layer which "compresses" all predicted flow vectors to produce a single transformation $(\hat{\mathbf{R}}, \hat{\mathbf{t}})$. This compression means that there could be significant noise in the individual flow vectors, even if they combine to form a reasonable action. Thus, we propose a consistency loss $L_{\text{consistency}}$ which serves as a regularizer to ensure that the *predicted* flow vectors are similar to their *induced, SVD-projected* flow vectors produced from the transformation encoded in $\hat{\mathbf{a}}$. The loss is:

$$L_{\text{consistency}}(\mathbf{P}, \hat{\mathbf{a}}) = \frac{1}{N'} \sum_{i=1}^{N'} \|(\hat{\mathbf{R}}p^{(i)} + \hat{\mathbf{t}} - p^{(i)}) - f^{(i)}\|_2, \tag{4}$$

where for each of the $N'$ tool points, we compute $\hat{\mathbf{R}}p^{(i)} + \hat{\mathbf{t}} - p^{(i)}$ as the induced flow from the predicted transformation $(\hat{\mathbf{R}}, \hat{\mathbf{t}})$ after applying the SVD layer, and $f^{(i)}$ is the flow predicted by the network before the SVD layer. Note that the ground truth transformation $(\mathbf{R}^*, \mathbf{t}^*)$ does *not* appear in this consistency loss. The consistency loss is only a function of a set of points and a set of corresponding flow vectors on those points, and does not rely on any other ground truth signal. We combine this with the point matching loss $L_{\text{point}}$ to obtain the final loss function to optimize the policy $\pi_\theta$: $L_{\text{combo}} = L_{\text{point}} + \lambda \cdot L_{\text{consistency}}$, with hyperparameter $\lambda$ controlling the weight of the consistency loss, which we set to $\lambda = 0.1$. See Figure 3 for visuals. To distinguish our method from traditional optical flow and scene flow methods, ToolFlowNet uses flow as a representation to compute the transformation of the tool, and is trained using the ground-truth demonstration action. It is not used to just estimate the flow.

**Additional Implementation Details:** To obtain ground truth tool flow $\mathbf{F}_{\text{gt}}$ in simulation, we determine the 3D movement of each tool point as a result of applying the demonstrator's action to transform those points. In physical settings, we scan the tool to obtain a 3D model, from which we extract tool point clouds $\mathbf{P}_{\text{tool}}$. We perform a similar calculation where we detect the transformation executed by the robot and apply it to obtain the flow for each tool point. This method of extracting $\mathbf{F}_{\text{gt}}$ only requires access to the current observed point cloud and the corresponding action. In particular, it does *not* require the perhaps more restrictive assumption of requiring one-to-one point correspondence between two consecutive point cloud observations. In addition, this method to detect flow means that it reflects the "intended" action from the robot, which may differ from the true positions of the tool points in 3D space after the robot executes the action; for example, when a collision happens with a wall, the tool points might not move, even though the robot intended for them to move. We leave alternative techniques to extract tool flow to future work.

## 4  Experiments

### 4.1  Simulation Experiments

We build on top of SoftGym [58], which provides a suite of deformable manipluation tasks and uses NVIDIA FleX [59] as the underlying physics engine. We use the simulator to obtain ground-truth segmentation labels. For the tool, we use the "observable" point cloud at each time step, so there may be occlusions. We test two tool-based simulation tasks, PourWater and ScoopBall, and for each, test two action spaces: 3D and 6D for PourWater, and 4D and 6D for ScoopBall. In PourWater, the agent

| Method | Loss | Dense PN++? | N. Success ScoopBall 4D | N. Success ScoopBall 6D | N. Success PourWater 3D | N. Success PourWater 6D | Average N. Success |
|---|---|---|---|---|---|---|---|
| PCL Direct Vector | MSE | ✗ | 0.544±0.03 | 0.848±0.05 | 0.530±0.08 | 0.402±0.04 | 0.581 |
| PCL Direct Vector | PM | ✗ | 0.228±0.12 | 0.048±0.04 | 0.132±0.07 | 0.088±0.04 | 0.124 |
| PCL Dense Transformation | MSE | ✓ | 0.519±0.07 | 0.824±0.06 | 0.539±0.05 | 0.344±0.03 | 0.556 |
| PCL Dense Transformation | PM | ✓ | 0.367±0.07 | 0.360±0.10 | 0.583±0.03 | 0.049±0.02 | 0.340 |
| D Direct Vector | MSE | ✗ | 0.190±0.07 | **0.952±0.02** | 0.035±0.01 | 0.069±0.02 | 0.311 |
| D+S Direct Vector | MSE | ✗ | 0.734±0.11 | **0.928±0.03** | **0.777±0.03** | 0.304±0.03 | 0.686 |
| RGB Direct Vector | MSE | ✗ | 0.354±0.05 | 0.776±0.05 | 0.698±0.02 | 0.324±0.05 | 0.538 |
| RGB+S Direct Vector | MSE | ✗ | 0.671±0.07 | **0.944±0.02** | **0.804±0.04** | 0.353±0.03 | 0.693 |
| RGBD Direct Vector | MSE | ✗ | 0.418±0.10 | 0.920±0.02 | 0.733±0.07 | 0.353±0.02 | 0.606 |
| RGBD+S Direct Vector | MSE | ✗ | 0.734±0.10 | **0.968±0.02** | **0.830±0.03** | 0.481±0.03 | 0.753 |
| ToolFlowNet, No Skip Conn. | PM+C | ✓ | 0.987±0.08 | 0.304±0.06 | 0.000±0.00 | 0.000±0.00 | 0.323 |
| ToolFlowNet, MSE after SVD | MSE+C | ✓ | 0.089±0.04 | 0.792±0.09 | 0.494±0.02 | **0.913±0.05** | 0.572 |
| ToolFlowNet, PM before SVD | PM | ✓ | 0.785±0.08 | 0.880±0.05 | 0.618±0.04 | 0.677±0.05 | 0.740 |
| ToolFlowNet, No Consistency | PM | ✓ | 0.861±0.06 | 0.744±0.12 | 0.468±0.10 | 0.609±0.06 | 0.670 |
| **ToolFlowNet (Ours)** | PM+C | ✓ | **1.152±0.07** | **0.952±0.02** | **0.795±0.05** | 0.667±0.03 | **0.892** |

Table 1: Behavioral Cloning (BC) results in simulation. The first 10 rows are baselines, the next 4 are ablations of our method, and the last row is our method. We report the loss functions used as MSE only, PM only (the loss in Eq. 3), or if it also uses a consistency loss (+C, from Eq. 4). We also show whether the method uses a segmentation PointNet++ (i.e., a dense architecture), and the *normalized* success rates (N. Success) across all tasks over 5 independent BC runs. The last column averages the success across the four columns. We bold the best numbers in the columns, plus any with overlapping standard errors.

controls a box which contains water and must pour the water into a fixed target box. In ScoopBall, the agent controls a ladle and needs to scoop a ball. See Appendix A.1 for more details.

### 4.1.1 Baseline Methods

We compare the proposed method with the following baselines (see Section 4.3 for ablations):

- **PCL Direct Vector**. Uses a *classification* PointNet++ network to directly estimate a vector action (with a translation and an axis-angle rotation). We test two variants, one which supervises with the MSE loss and another which uses the Point Matching (PM) loss from Eq. 3 on tool points.
- **PCL Dense Transformation**. Uses a *segmentation* PointNet++, and directly predicts per-point 6D vectors (translation and axis-angle). Each point cloud has a designated point as the center of rotation for the tool, and we use the output corresponding to that point as the vector action. The outputs for the other points are not supervised. This baseline is designed to isolate any benefits from using the segmentation version of PointNet++ instead of classification. As with Direct Vector, we test two variants, with supervising using the MSE or PM losses.
- **{D, RGB, RGBD} Direct Vector**. Processes images and uses a Convolutional Neural Network to directly predict an action vector (translation and axis-angle) and supervises with MSE. The inputs are either a depth image (D), the RGB image, or an RGBD image.
- **{D+S, RGB+S, RGBD+S} Direct Vector**. These are the same as the prior set of methods, except that the input images have additional channels corresponding to binary segmentation masks. We denote these new input images as: D+S, RGB+S, and RGBD+S. We include these baselines for a fairer comparison due to assuming segmentation information in the point cloud observations.

### 4.1.2 Experiment Protocol and Evaluation

For each task, we use a scripted demonstrator to generate a fixed set of training demos and keep the successful ones for Behavioral Cloning. We standardize on 500 training epochs for all methods and average across 5 seeds. We evaluate every 25 training epochs on 25 testing configurations and use the maximum success (averaged over 5 seeds) across the full training history, then divide this by the demonstrator success rate to get the normalized performance. See Appendix A.2.2 for more details.

### 4.2 Simulation Results and Analysis

The results in Table 1 suggest that using ToolFlowNet outperforms the baselines on average across the tasks. In particular, for ScoopBall 4D and PourWater 6D, it outperforms all other baselines, and

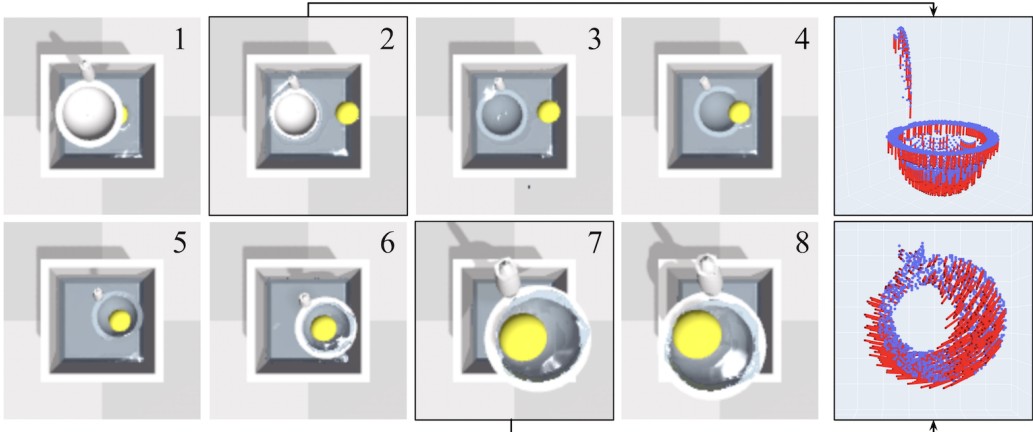

Figure 4: An example successful ScoopBall 4D rollout by a learned ToolFlowNet Behavioral Cloning policy. We show subsampled RGB frames for visual clarity, though the policy only uses point clouds as input. For two of the frames, we show the policy's tool flow visualizations. The policy lowers the ladle (frames 1-3), rotates and moves it in the direction of the ball (frames 4-5), lifts the ball (frame 6) and then rotates back to the starting pose (frames 7-8). The policy's flow visualizations for frames 2 and 7 suggest the ability to learn downward and rotation movement, respectively. The predicted flow vectors, colored red, are slightly enlarged for clarity.

for ScoopBall 6D and PourWater 3D, it is on par with the best image-based baselines. This may indicate that some tasks have a 3D nature which makes it more natural to learn policies from point clouds. Figure 4 shows a qualitative example test-time rollout of ScoopBall 4D from the learned ToolFlowNet policy. Figure 4 also visualizes the policy's internal flow predictions (i.e., the per-point flow vectors $f^{(i)}$ before the SVD layer), showing that the network has learned surprisingly clean per-point tool flow vectors. Furthermore, as the agent controls the ladle at its upper tip, when rotating, the flow vectors also correctly predict longer flow vectors for the points located further away from the origin of the tool pose.

### 4.3 Why Does ToolFlowNet Help Robot Learning?

We perform further experiments to determine why ToolFlowNet outperforms the baselines that directly regress to a transformation. Specifically, we create a variant of ScoopBall in which the action space consists of translations only (no rotations); see Appendix B.1 for details. These experiments reveal that ToolFlowNet does not outperform the baselines in translation-only settings, indicating that the benefits of ToolFlowNet come from predicting rotations. We also test Direct Vector methods with 4D (quaternions), 6D [60], 9D (rotation matrices) [54], and 10D [61] rotation representations in Appendix B.12, and find that ToolFlowNet continues to obtain higher success rates.

We next study ablations of ToolFlowNet to understand which components are most critical:

- **ToolFlowNet, No Skip Connections**: removes skip connections in the segmentation PointNet++.
- **ToolFlowNet, MSE after SVD**: tests applying an MSE loss on the induced transformation from SVD instead of point matching. We still use the consistency loss (Eq. 4).
- **ToolFlowNet, Point Matching (PM) Before SVD**: tests using the PM loss (Eq. 3) before the SVD layer, so the loss does not back-propagate through the SVD layer.
- **ToolFlowNet, No Consistency**: tests removing the consistency loss (and just using Eq. 3).

We use the same experiment protocol as in Section 4.1.2 on all tasks. The results, also in Table 1, suggest strong benefits to using the point matching loss, the consistency loss, and the standard segmentation PointNet++ with skip connections. For example, across all tasks, ToolFlowNet performance is worse without using a consistency loss. The utility of some design choices may be more task-specific; removing skip connections leads to no successes on PourWater because removing it made the model unable to predict any rotations (see Appendix B.4 for additional analysis), while it is possible to succeed in ScoopBall without using rotations.

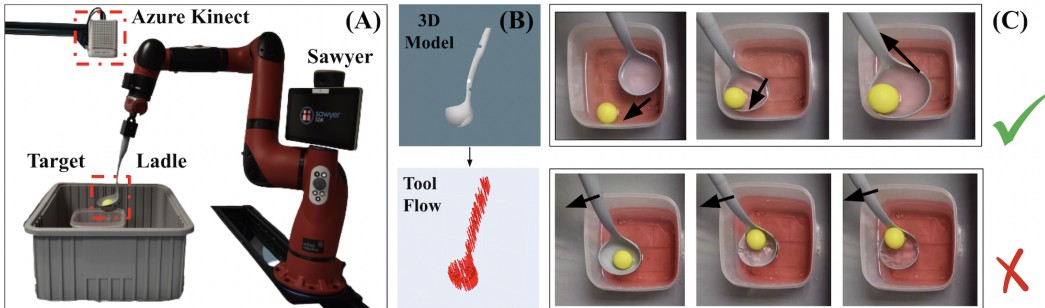

Figure 5: Physical experiments. (A) The Sawyer holds a ladle above a small box with water, which is enclosed in a larger gray box to contain spills. (B) The scanned 3D model of the ladle with a representative tool flow visualization. (C) Example test-time trials with subsampled frames. Top row: successful tool movement towards and lifting the target. Bottom row: collision failure due to repeatedly pushing against the container.

## 4.4 Physical Scooping Experiments

We test ToolFlowNet in the real world using a Sawyer robot with a standard consumer ladle which we scan to obtain a 3D model, and a yellow ping-pong ball acting as the target item (see Figure 5). The ladle is attached to the Sawyer's end-effector. As in simulation, we obtain tool flow by recording the change in end-effector pose and applying the transformations on the tool point cloud. A Microsoft Azure Kinect camera captures top-down depth images to compute the ball's point cloud.

A human operator manually moves the Sawyer's arm via direct touch to collect 125 demonstrations, with each comprising about 20 observation-action pairs. We use 100 demonstrations for training ToolFlowNet, and use the remaining 25 for monitoring evaluation MSE. We perform 50 physical scooping trials, where each trial begins with the human dropping the ping-pong ball over the water at an

| ToolFlowNet in Real | #Trials |
|---|---|
| Successes | 41/50 |
| Failures | 9/50 |

Table 2: Physical scooping results.

arbitrary location within the inner box shown in Figure 5. The trial is classified as successful if the Sawyer raises the ball from the water surface to above the top of the smaller box. Results in Table 2 suggest that the Sawyer achieves 41/50 successes (82%), with 9 failures. All failures were due to the ladle colliding with the small box. See Appendix C for more details. In future work we will do physical experiments with more complex demonstrations.

## 4.5 Limitations and Failure Cases

In our experiments, we obtain the ground-truth tool flow data by applying the demonstrator's actions on a set of tool points and computing the change in the resulting tool point positions. The tool points can be observed or derived via a tool model. In either case, we require access to the demonstrator's action, and future work could relax this assumption by extracting tool flow without explicit actions, such as when a human provides a video. Possible approaches include using scene flow techniques.

A limitation of ToolFlowNet is that it may be susceptible to occlusions of the tool when a model of the tool is not available. For physical scooping, we rely on a scanned model of the tool because the Sawyer's arm would occlude parts of the tool, but tool models might not always be available. In future work, we will explore ways to address occlusions such as point cloud inpainting and tracking. Finally, we test ToolFlowNet on two simulation tasks with two action spaces for each, and scooping in real. We hope to test on more diverse tasks such as cloth or rope manipulation, and to address failures from the physical experiments.

## 5 Conclusion

In this work, we propose a general technique for policy learning from point clouds for tool-based manipulation tasks, which we demonstrate on scooping and pouring tasks. Our method, called ToolFlowNet, predicts per-point tool flow vectors which are converted into actions. We hope this research inspires future work on learning from point clouds.

**Acknowledgments**

This work was supported by LG Electronics and by NSF CAREER grant IIS-2046491. We thank Brian Okorn and Chuer Pan for assistance with the differentiable SVD layer, and Mansi Agrawal, Sashank Tirumala, and Thomas Weng for paper writing feedback.

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
