# OpenReview forum: "ToolFlowNet: Robotic Manipulation with Tools via Predicting Tool Flow from Point Clouds"
_robot-learning.org/CoRL/2022/Conference — CoRL 2022 Poster_

### Official Review · Reviewer_Vzsj · 2022-07-11

**Originality:** Very Good
**Technical Quality:** Very Good
**Clarity Of Presentation:** Very Good
**Impact:** 4

**Recommendation:**

Weak Accept: I recommend accepting the paper, but will not argue for my recommendation if the majority of other reviewers have a different opinion.

**Summary:**

The paper proposes ToolFlowNet, a novel point-cloud-based architecture that predicts the per-point tool flow vectors. Those vectors are used to formulate an SE(3) transformation as an action for the robot to execute. Experimental evaluation is performed in both the simulation and the real world with convincing results.

**Issues:**

The Image CNN baselines only process RGB information but not depth information, which seems a bit unfair.

A minor question: in the real-world experiment, how do you obtain the segmentation labels?

**Quality Of The Limitations Section:**

Limitations are addressed clearly

**Reviewer Expertise:**

4: The reviewer is confident but not absolutely certain that the evaluation is correct

**Robotics Focus:**

Sufficient demonstration on hardware

**Strengths And Weaknesses:**

Strengths:

1. The idea of using per-point flow vectors to formulate an action is novel and interesting. Though [33] used a similar idea to predict articulation flow for manipulating articulated objects, this paper proposes a novel perspective to use flow vector to control an arbitrary tool.
2. The paper is well-written and easy to follow.

Weaknesses:

1. The Image CNN baseline seems unfair because it uses RGB images instead of RGBD images or depth images while the proposed method uses point cloud that contains depth information.
2. Although the method is defined in SE(3), the action space in the experiments is limited in 3D ($\Delta x$, $\Delta y$, $\Delta \theta$) or 4D ($\Delta x$, $\Delta y$, $\Delta z$, $\Delta \theta$) instead of 6D. The limitation section does not discuss why 6D task is not demonstrated and the potential issues when using this method in a 6D action space.
3. Showing two tasks in Figure 2 is a bit misleading. I thought the module takes in two separate point clouds at first glance. I would recommend only keeping one task in this figure.

**Summary Of Recommendation:**

The paper proposes a novel approach to predict the per-point flow using a point net architecture, then combine the per-point flow on a tool to generate an SE(3) action of the end-effector. Though the paper has some minor weaknesses (see above), I think it is a good contribution that provides new insight on how to use point cloud and point net architectures in robotic manipulation.

---

> ### Author Response · Authors · 2022-08-27
> **Response to Reviewer Vzsj**
>
> Dear Reviewer,
>
> We thank you for your feedback and for the positive review. We have revised the paper draft based on your suggestions with changes highlighted and have included new experiments. We have attached the revised PDF to this post. Below, we respond to your comments:
>
> > **Q1**: The Image CNN baseline seems unfair because it uses RGB images instead of RGBD images or depth images while the proposed method uses point cloud that contains depth information.
>
> **A1**: As suggested, in the revised version of the paper, we present new comparisons to **methods that learn from depth images, RGB images, and RGBD images**. In addition, because ToolFlowNet relies on segmented point clouds as inputs, to further strengthen the image-based baselines, **we also compute the corresponding binary segmentation mask images** and provide that as input by stacking them as additional image channels. Table 1 and Section 4.1.1 have additional details to clarify these points, and Section A.2.3 in the Supplementary Material describes more details of the (task-specific) segmentation masks. **We find that ToolFlowNet is still able to achieve higher success rates on average** compared to these different baselines which use depth, RGB, or RGBD images as input (even when they have segmentation masks).
>
> > **Q2**: The limitation section does not discuss why 6D task is not demonstrated and the potential issues when using this method in a 6D action space.
>
> **A2**: We agree that we should evaluate our method on tasks with a 6D action space. **We have performed an additional set of experiments for both PourWater and ScoopBall that consider 6 DoF action spaces**. We have updated Table 1 in the paper with quantitative imitation learning results for the 6 DoF variants. Overall, **the results are consistent with showing the benefits of ToolFlowNet** for learning these tasks. The proposed ToolFlowNet model continues to outperform baselines as well as ablations on average. We have also updated **Section B.4** in the supplement with more quantitative analysis, and have updated the project website (https://tinyurl.com/toolflownet) for qualitative results, with more visualizations and GIFs of these environments and action spaces in the subsection titled "**[Simulation Experiments](https://sites.google.com/view/point-cloud-policy/home#h.bnb6liddj0n2)**".
>
> > **Q3**: Showing two tasks in Figure 2 is a bit misleading. I thought the module takes in two separate point clouds at first glance. I would recommend only keeping one task in this figure.
>
> **A3**: We agree with this suggestion and have adjusted Figure 2 in the paper (as well as added more notation to make it better match the text).
>
> > **Q4**: A minor question: in the real-world experiment, how do you obtain the segmentation labels?
>
> **A4**: For the physical experiments, we continuously query RGB and depth images from an overhead Azure Kinect Camera, from which we generate point clouds of the scene. Given the distinct yellow color of the ping-pong ball, we can segment out the ball points at each time step, using HSV thresholding on the RGB images. We use the resulting HSV mask to filter out only the target points from the observed point cloud. For the tool, however, we make use of a known 3D model of the tool. At each time step, we can query the Sawyer’s end-effector pose, and combined with the knowledge of the 3D model of the tool, we can analytically compute the pose for a set of points for the tool point cloud. We sample points from this 3D model of the tool to obtain the tool points at a given time-step. Through this technique, we obtain segmentation classes for both the target and the tool points. We have modified Section C.1 in the Supplementary Material to provide additional clarity.
>
> _Once again, thank you for your suggestions. Please let us know if you have any further clarifications that you think we need to address._

---

### Official Review · Reviewer_ckzp · 2022-07-18

**Originality:** Good
**Technical Quality:** Good
**Clarity Of Presentation:** Fair
**Impact:** 2

**Recommendation:**

Weak Accept: I recommend accepting the paper, but will not argue for my recommendation if the majority of other reviewers have a different opinion.

**Summary:**

This paper resents a method in which robust policies can be learned using only few demonstrations. It does this on the basis of point cloud inputs and a network that outputs the predicted future optical flow according from a set of given scripted demonstrations. The paper includes extensive baseline experiments in combination with real world experiments. The paper is well written but the method is difficult to understand.

**Issues:**

Major Points:
The method is not explained in a very clear way. My current understanding is as follows:

a) The policy a = \pi(o) computes the dense flow F as an intermediate step, so a = \pi ( F(o) ). This is, however, not the flow in the classical sense t_{-1} -> t_0, but the predicted flow into the future. From F we then extract T, which we convert into a:  a = \pi(T(F_\theta(o))). The network producing F is then supervised using only a. An explicit evaluation of F, on new inputs, is not possible, since it depends not only on the tool positions, but also the manipulated object e.g. yellow ball position.

a2) I was then confused when I read the limitations section, I initially thought ground truth tool flow refers to F, but it probably means T.

b) For the In the DirectVector MSE baseline its not clear if MSE is applied to the actions directly or the points, both variants should be included.

c) The notation is a bit unclear. Input point clouds are always referred to as p, it is introduced as *segmented* point clouds. I am assuming that the point clouds required during the runtime of the algorithm do not need to be segmented. This is a bit confusing in combination with the references to "classification" and "segmentation" PointNets in L195, L199, as L195 seems to mean an action regression network and L199 a per-point regression network. The potential confusion is wrt. discrete action classification and tool mask segmentation.

Why is the method difficult to understand:
a) Someone coming from an optical flow or scene flow background would expect a ToolFlowNet to output F, and a network to be trained and evaluated on this task. This difference should be more explicitly highlighted.

b) Understanding what the method did required piecing together information from different sections, more explicit formulas like a = \pi(T(F_\theta(o))) would help.

c) Some variables in Figure 2. would also help.

A reference to the "FlowControl: Optical Flow Based Visual Servoing" should be included, as indicated by the name many aspects of the paper are similar to this work, also addressing the topic of robot manipulation using segmented RGB-D inputs to compute MSE between point clouds to find relative transformations for robot motion.  Maybe also "Coarse-to-Fine Imitation Learning: Robot Manipulation from a Single Demonstration"

Minor Points:

Why not include color information in the point clouds, you should get this for "free" in the simulation as well as your experimental setup.

The Point matching loss solves a similar objective to the MSE loss, having these appear explicit side by side would clarify the differences.

In line 115 there should probably be a P'_{tool},

Line 115/120 seem to be a bit redundant, maybe it could be reformulated so that T*,F_gt,F,a*,a^ are explained compactly as these are  similar.

**Quality Of The Limitations Section:**

Limitations are not well addressed

**Reviewer Expertise:**

4: The reviewer is confident but not absolutely certain that the evaluation is correct

**Robotics Focus:**

Sufficient demonstration on hardware

**Strengths And Weaknesses:**

Strengths:

1. Presents a method in which robust policies can be learned using only few demonstrations.
2. The paper combines a relatively extensive ablation study for simulated results
3. The paper performs real works experiments.

Weaknesses:

1. The method is not easily understood.
2. Some references that should be included are missing.

**Summary Of Recommendation:**

I think that the paper presents a good method, with good experiments. However it seems to be hastily written and is still lacking in clarity, substantial revisions should be made before acceptance.

---

> ### Author Response · Authors · 2022-08-27
> **Response to Reviewer ckzp (Part 1 of 2)**
>
> Dear Reviewer,
>
> We thank you for your time and for the paper feedback. We sincerely appreciate your suggestions to improve clarity and we apologize for any confusion. We have revised the paper draft based on your suggestions with changes highlighted. We have attached the revised PDF to this post. Below, we respond to your questions:
>
> > **Q1**: “... From F we then extract T, which we convert into a: a = \pi(T(F_\theta(o))) …  An explicit evaluation of F, on new inputs, is not possible, since it depends not only on the tool positions, but also the manipulated object e.g. yellow ball position.”
>
> **A1**: Thank you for bringing up these two points. To clarify, in our paper, as noted in the second paragraph of Section 3.1, the action $\mathbf{a}$ is the transformation $\mathbf{T}$. Therefore, the action can be written as $\mathbf{a} = \mathbf{T} = \pi(\mathbf{o})$ where in the paper we use $\pi$ to refer to the end-to-end ToolFlowNet policy which, given an input observation $\mathbf{o}$ produces the action $\mathbf{a}$. In this paper, the observations are segmented point clouds $\mathbf{P}$, elaborated later in our response (see answer A4). The policy $\pi$ first processes this observation input through a segmentation PointNet++ architecture (which we denote as $F_\theta$) which produces per-point estimated flow denoted as $\mathbf{F}$. Then a differentiable SVD layer takes this estimated (tool) flow $\mathbf{F}_{\rm tool}$ (a subset of $\mathbf{F}$) to extract $\mathbf{a}$. The SVD layer has no parameters, so we also use $\theta$ to refer to the policy parameters $\pi_\theta$.
>
> Second, when estimating the tool flow, the input to the ToolFlowNet policy $\pi$ includes both the tool points and the points from the manipulated object, e.g., the yellow ball in ScoopBall. Therefore, it is possible to estimate flow on new inputs and the estimated flow can take into consideration the position of the manipulated object. We have updated Sections 3.1 and 3.2 and added Equation 1 to clarify these points. Please let us know if anything is still unclear.
>
> > **Q2**:  I was then confused when I read the limitations section, I initially thought ground truth tool flow refers to F, but it probably means T.
>
> **A2**: Thank you for your question. To clarify, we refer to the ground-truth tool flow as $\mathbf{F}_{\rm gt}$, which is computed by applying the ground-truth transformation $\mathbf{T}^{*}$ on the tool points:
>
> $\mathbf{F}_\textrm{gt} = \mathbf{T^*}\mathbf{P}_\textrm{tool} - \mathbf{P}_\textrm{tool}$.
>
> We use the notation $\mathbf{F}$ to refer to the per-point predicted flow from the segmentation PointNet++. We have updated the second paragraph of Section 3.1 to clarify this point and (as in our answer A7) we have added notation to Figure 2.
>
> > **Q3**: For the In the DirectVector MSE baseline its not clear if MSE is applied to the actions directly or the points, both variants should be included.
>
> **A3**: Thank you for your question. To clarify, **Direct Vector MSE** means the MSE loss is applied on the estimated transformation (translation + rotation), written as a vector. We also have a variant of applying MSE directly on the (tool) points based on the transformation computed from the tool flow after performing SVD, which is the **Direct Vector PM** (point matching) variant in the paper. We report performance **of both variants in the paper**. We have updated Section 4.1.1 to clarify this point.
>
> > **Q4**: The notation is a bit unclear. Input point clouds are always referred to as p, it is introduced as segmented point clouds. I am assuming that the point clouds required during the runtime of the algorithm do not need to be segmented. This is a bit confusing in combination with the references to "classification" and "segmentation" PointNets in L195, L199, as L195 seems to mean an action regression network and L199 a per-point regression network. The potential confusion is wrt. discrete action classification and tool mask segmentation.
>
> **A4**: Thank you for pointing this out. In this paper all point cloud observations are assumed to be “segmented” in that they contain, for each point, both the XYZ position and a one-hot vector encoding representing the class of that point. ScoopBall has two classes: the tool (ladle) and the ball. PourWater has three classes: the tool (controlled box), the fixed target box, and the water. This is the case both for the offline demonstration data *and* for the point clouds during evaluation (runtime). The naming of these PointNet++ networks is due to the task that they were originally designed for. In this paper, they are not used for discrete action classification or tool mask segmentation. We have updated Section 3.1 to clarify that point cloud inputs are already segmented.

---

> ### Author Response · Authors · 2022-08-27
> **Response to Reviewer ckzp (Part 2 of 2)**
>
> > **Q5**: Someone coming from an optical flow or scene flow background would expect a ToolFlowNet to output F, and a network to be trained and evaluated on this task. This difference should be more explicitly highlighted.
>
> **A5**: ToolFlowNet uses flow as an intermediate representation to compute the final transformation of the tool, and is trained using the demonstration action. We have added more clarifications on this in the updated Section 3.2.
>
> > **Q6**: Understanding what the method did required piecing together information from different sections, more explicit formulas like a = \pi(T(F_\theta(o))) would help.
>
> **A6**: Thank you for the suggestion! We have added more explicit formulas at the end of Section 3.1, in Equation 1.
>
> > **Q7**: Some variables in Figure 2. would also help.
>
> **A7**: Thank you for this suggestion. We have updated Figure 2 with variables.
>
> > **Q8**: A reference to the "FlowControl: Optical Flow Based Visual Servoing" should be included, as indicated by the name many aspects of the paper are similar to this work, also addressing the topic of robot manipulation using segmented RGB-D inputs to compute MSE between point clouds to find relative transformations for robot motion. Maybe also "Coarse-to-Fine Imitation Learning: Robot Manipulation from a Single Demonstration"
>
> **A8**: We thank the reviewer for pointing out these two related works. Indeed, FlowControl is a highly relevant paper that we should have cited. In the revised paper, we added a discussion of FlowControl in the Related Works (Section 2), and we cite “Coarse-to-Fine Imitation Learning” at the beginning of Section 3.
>
> > **Q9**: Why not include color information in the point clouds?
>
> **A9**: We thank the reviewer for this interesting idea. We did not include color information because it is not necessary for the tasks; both PourWater and ScoopBall can be completed using the position and class label of the point in the point clouds; we mention the class labels in our response to Q4. Further, we are worried that including color information would hurt generalization to new scenarios in which the object color or scene lighting might change.  However, it is possible to include color information as additional channels of the input point cloud to ToolFlowNet, and we leave this for future work.
>
> > **Q10**: The Point matching loss solves a similar objective to the MSE loss, having these appear explicit side by side would clarify the differences.
>
> **A10**: We thank the reviewer for this suggestion on improving the clarity of the paper. We have added the equation of the MSE loss in Section 3.2, Equation 2, just before Equation 3 which shows the Point Matching loss.
>
> > **Q11**: In line 115 there should probably be a P'_{tool},
>
> **A11**: Thank you for the suggestion. We define $\mathbf{P}’_\mathrm{tool}$ a few lines later (line 128 in the revised version), which is the result of moving each point in $\mathbf{P}_\mathrm{tool}$ by the estimated flow vectors $\mathbf{F}_\mathrm{tool}$. We use $\mathbf{T}^* \mathbf{P}_\mathrm{tool}$ in line 115 of the submitted version (122 in the revised version) to denote moving the entire point cloud $\mathbf{P}_\mathrm{tool}$ by the transformation $\mathbf{T}^*$ from the demonstration. We have updated Section 3.1 to clarify this point.
>
> > **Q12**: Line 115/120 seem to be a bit redundant, maybe it could be reformulated so that T*,F_gt,F,a*,a^ are explained compactly as these are similar.
>
> **A12**: We thank the reviewer for this suggestion. We agree that it would help to present these concepts compactly, and have included Equation 1 to clarify this point (as per A6).
>
> > **Q13**: Quality Of The Limitations Section: Limitations are not well addressed
>
> **A13**: We are happy to add more information to the limitations section, if the reviewer could let us know the areas we should expand on.
>
> _Once again, thank you for your suggestions. We greatly appreciate your feedback. Please let us know if you have any further clarifications that you think we should address._

---

### Official Review · Reviewer_sFXa · 2022-07-25

**Originality:** Good
**Technical Quality:** Fair
**Clarity Of Presentation:** Good
**Impact:** 2

**Recommendation:**

Weak Reject: I recommend rejecting the paper, but will not argue for my recommendation if the majority of other reviewers have a different opinion.

**Summary:**

The paper proposes a method to learn behavioral cloning motion policies for arbitrary tools, given a point cloud.  In particular, the network introduces a network architecture to do so.

The main contributions of the paper are:
1. A network architecture that learns the tool’s motion policies through the learning of the point cloud flow. The authors claim that estimating
    the tool’s point cloud flow F prior to the tool’s transformation T leads to improved performance with respect to estimating the tool’s
    transformation directly.
2. The paper is evaluated in two simulated tasks. Learning how to pour water from a box and learning how to scoop a ball with a ladle.
3. A real robot experiment in which the scoop of a ball is learned from real human demonstrations.


**Issues:**

Major issues has been previously introduced.

Minor:
In line 123, T should be bold.




**Quality Of The Limitations Section:**

Additional details required

**Reviewer Expertise:**

5: The reviewer is absolutely certain that the evaluation is correct and very familiar with the relevant literature

**Robotics Focus:**

Highly relevant to robotics but no hardware experiments

**Strengths And Weaknesses:**

To the reviewer's view, the main strengths of the paper are:

1. A fully differentiable architecture that outputs the translation vector and rotation matrix of the tool, through the tool’s point cloud flow. I was not aware that it was possible to fully differentiate the whole pipeline.

2. Considering the whole scene point cloud as input observation data to learn the tool’s motion policy.

3. If sufficiently validated, the claim that “estimating the tool flow leads to improved performance compared to estimating the transformation T directly”.

To the reviewer's view, the main weaknesses of the paper are:

1. Too simple experiments. There is no experiment considering 6D movements. The chosen experiments assume: 2D translation + 1D rotation in the pouring experiment, 3D translation + 1D rotation in the ball scooping and 3D translation for the physical scooping experiment. To properly validated the proposed method, it is essential to show its performance in a 6D motion generation task. If a motion is not exactly aligned with the world frame, the motion representation is going to be decoupled along multiple axis and thus, highly beneficial to show that the system is able to learn these motions both with simulated and real data.

2. The paper misses a proper analysis on why  “estimating the tool flow leads to improved performance compared to estimating the transformation T directly”. It is not intuitive why learning the flow of a whole pointcloud should enhance the performance. Previous works on learning behavioral cloning motion policies on SE(3) [1, 2, 3, 4] directly learn the motion policy in SE(3), without the pointcloud flow. These works assume access to the object’s 6D pose and learn the change in the pose. It would be relevant to observe if the pointcloud flow based motion representation is actually better than first computing the 6D pose of the object and then, estimating the transformation given the pose. This would help understanding the benefit of pointcloud flow.

3. The network is trained using the flow of the tool’s pointcloud. Therefore, the data is required to store a one-to-one mapping between all the points in the pointcloud. Yet, when recording the pointcloud with and RGB-D camera, this information is missing or some regions of the tool might be occluded in some frames. It would be interesting to know if the method is limited to pointcloud data that has been generated synthetically or if the authors have any proposition on how to extend the method to raw RGB-D training data in which the alignment between the points is missing. If the solution is to estimate first the pose, then, could we use this pose as proposed in weakness 2?

[1] Pastor, Peter, et al. "Online movement adaptation based on previous sensor experiences." 2011 IEEE/RSJ International Conference on Intelligent Robots and Systems. IEEE, 2011.

[2] Huang, Yanlong, et al. "Toward orientation learning and adaptation in cartesian space." IEEE Transactions on Robotics 37.1 (2020): 82-98.

[3] Rozo, Leonel, and Vedant Dave. "Orientation Probabilistic Movement Primitives on Riemannian Manifolds." Conference on Robot Learning. PMLR, 2022.

[4] Urain, Julen, Davide Tateo, and Jan Peters. "Learning Stable Vector Fields on Lie Groups." arXiv preprint arXiv:2110.11774 (2021).

**Summary Of Recommendation:**

The paper presents an idea that could be interesting for learning motion policies from point clouds. Nevertheless, the work is missing a proper experimental validation and a proper comparison with respect to classic methods (Compute the pose of the object and learn the motion given the known pose).

The authors need to show the performance of their proposed method for more complex tasks that require 6D motion.

The authors need to show the difference in using the point cloud flow with respect to estimating the tools pose and using this state information to learn the motion policy (classic approach) and thus, validate their claim that “estimating the tool flow leads to improved performance compared to estimating the transformation T directly”.

The work would also benefit from an experiment showing the performance of the method with real RGB-D data instead of synthetic one.

Due to these limitations, even if the work has potential, I suggest a weak rejection of the paper until the arisen problems are tackled.

---

> ### Author Response · Authors · 2022-08-28
> **Response to Reviewer sFXa (Part 1 of 2)**
>
> Dear Reviewer,
>
> We thank you for your time and for the paper feedback. We have revised the paper draft based on your suggestions with changes highlighted and have included new experiments. We have attached the revised PDF to this post. Below, we respond to your questions:
>
> > **Q1**: There is no experiment considering 6D movements.
>
> **A1**: We agree that this was a limitation of our experiments. In the revised version of the paper, we now have experiments on 6D action movements for both the **ScoopBall** and **PourWater** environments. We have repeated the full set of experiments for ToolFlowNet, ablations and other baselines on these environments. We find that ToolFlowNet still achieves the highest normalized success rates, on average, compared to alternative methods. The main results are in Table 1 and additional details are in the **Supplementary Material Section B.4**.
>
> We also have updated the project website which shows examples of full 6D movements for both ScoopBall and PourWater. You can see GIFs of the demonstrators in the section **[Demonstration Data and Flow Visualizations](https://sites.google.com/view/point-cloud-policy/home#h.xcag6otr00dl)**, and GIFs of the learned ToolFlowNet policy in the section **[ToolFlowNet: Learned Policy Rollouts](https://sites.google.com/view/point-cloud-policy/home#h.rf0vlv1gx48f)**. These include visualizations of the new 6D action spaces.
>
> > **Q2**: The paper misses a proper analysis on why “estimating the tool flow leads to improved performance compared to estimating the transformation T directly”. It is not intuitive why learning the flow of a whole pointcloud should enhance the performance. Previous works on learning behavioral cloning motion policies on SE(3) [1, 2, 3, 4] directly learn the motion policy in SE(3), without the pointcloud flow. These works assume access to the object’s 6D pose and learn the change in the pose. It would be relevant to observe if the pointcloud flow based motion representation is actually better than first computing the 6D pose of the object and then, estimating the transformation given the pose. This would help understanding the benefit of pointcloud flow.
>
> **A2**: Thank you for the helpful suggestions. We have performed additional experiments where **we assume access to ground-truth tool and object poses** and train a method to regress from the tool and object poses to the robot actions, using a simple MLP network. For ScoopBall the input is a concatenated vector of the ladle pose and the center of the ball. For PourWater the input is a concatenated vector of the poses of the controlled box and the target box, along with the fraction of water particles in each box. We call this method “State (G.T.) Direct Vector”.
>
> We present the state-based imitation learning results in **Supplementary Material Section B.6 (“State-Based Policy Baseline”)**. We find that State (G.T.) Direct Vector indeed performs well and matches ToolFlowNet on 2 of the 4 env and action variants, slightly outperforms it on 6D pouring, though it is much worse at 6D scooping. On average, ToolFlowNet gives somewhat better performance. Nonetheless, the state-based policies **require access to ground-truth state information**, which includes the poses of the objects in the environment, whereas ToolFlowNet does not require access to this ground-truth information.
>
> Thank you for pointing out the references [1,2,3,4]. We have added them as related works in the Supplement, Section B.6.
>
> We agree that it is helpful to better understand as to why learning the flow of the whole point cloud should enhance performance over regressing to a transformation. One hypothesis we have is that a set of flow vectors can be easier for a Deep Neural Network to predict than directly regressing to a rotation. We have explored this claim in the **Supplement, Section B.1.1** where we tested a 3D translation-only variant of the 4D ScoopBall environment. When the demonstrator only moves using 3D translations without rotations, ToolFlowNet performs slightly worse than directly regressing to the translation (with no rotation). Thus the benefit of ToolFlowNet seems to be when a rotation is involved; in such cases our method significantly outperforms directly regressing to the transformation.
>
> This point is further demonstrated by the prevalence of pose estimators in the literature which do not regress directly to rotations. For example, SurfEmb (https://surfemb.github.io/) is a state-of-the-art object pose estimator which achieves strong performance on the BOP Challenge for pose estimation (https://bop.felk.cvut.cz/home/), and SurfEmb estimates pointwise embeddings instead of directly regressing the object pose.

---

> ### Author Response · Authors · 2022-08-28
> **Response to Reviewer sFXa (Part 2 of 2)**
>
> > **Q3**: The network is trained using the flow of the tool’s pointcloud. Therefore, the data is required to store a one-to-one mapping between all the points in the pointcloud. Yet, when recording the pointcloud with and RGB-D camera, this information is missing or some regions of the tool might be occluded in some frames. [...]
>
> **A3**: Thank you for this question. To clarify: **ToolFlowNet does not require a 1:1 mapping between all points in the point cloud**. Having such information could indeed be used for training ToolFlowNet, but can be an overly restrictive assumption as you correctly imply. In the paper we only require that for each time step, we have the (segmented) tool point cloud and the robot action. Then, we can use the robot action to compute the transformation of all the points in the tool point cloud, to obtain the tool point cloud at the next time step. This automatically gives us the correspondence for tool points across frames. We did this for all the experiments (in simulation and in the real world) in the paper, showing that such an approach is feasible. In real experiments, we use a scanned model of the tool, so that its pose can be inferred at each time step by the pose of the robot end-effector. If a model of the tool is not available, then the observed tool points could also be used, and the robot action can still be used to transform the points and obtain the correspondence across frames. We use the observed point cloud in all simulation experiments in this paper. In Section 4.6 (limitations and future work), we discuss how it might be possible to learn from human demonstrations without having direct access to the demonstrator’s actions, such as by using a method to estimate the scene flow.
> We have added a clarification on this point in the paragraph “**Additional Implementation Details” (in Section 3.2)**.
>
> > **Q4**: The work would also benefit from an experiment showing the performance of the method with real RGB-D data instead of synthetic one.
>
> **A4**: To clarify, we have included physical scooping experiments **which make use of real RGB-D data**, which we report in Section 4.5 and in Appendix C.
>
> > **Q5**: Robotics Focus: Highly relevant to robotics but no hardware experiments
>
> **A5**: As mentioned above, **we have physical scooping experiments**, which we report in Section 4.5 and in Appendix C.
>
> > **Q6**: Minor: In line 123, T should be bold.
>
> **A6**: Thanks for noticing this typo. We have fixed this in the latest version.
>
> _We appreciate the helpful feedback. Please let us know if these address your concerns._

---

### Official Review · Reviewer_LESm · 2022-07-26

**Originality:** Good
**Technical Quality:** Very Good
**Clarity Of Presentation:** Very Good
**Impact:** 3

**Recommendation:**

Weak Accept: I recommend accepting the paper, but will not argue for my recommendation if the majority of other reviewers have a different opinion.

**Summary:**

This paper presents a framework for learning robotic manipulation tasks with tools using point clouds. To extract point cloud observations, it utilizes a dense point cloud network. It proposes a point-cloud-to-action system, ToolFlowNet, to generate a full transformation matrix to estimate the motion of point clouds. An algorithmic demonstrator along with the ToolFlowNet is used to generate a dataset for training a behavior cloning policy.

**Issues:**

Introduction:
* Consider re-writing the first sentence (lines 17 to 19). The first sentence of the introduction should be clear and concise.

Method:
* [3.2] It’s not clear why the ground truth transformation is not used in the consistency loss. Consider adding this information on line 154.

Experiments:
* Table 1: Why does “ToolFlowNet, No Skip Conn.” have a success of 0% in the PourWater task?  How would skip connections help when learning to predict rotations? The authors should explain this more clearly (lines 243 - 245).
* It is a good idea to show the success rate of the demonstration data in Table 1.


**References:**

[1] Bokui Shen, Zhenyu Jiang, Christopher Choy, Leonidas J. Guibas, Silvio Savarese, Anima Anandkumar, and Yuke Zhu. ACID: Action-Conditional Implicit Visual Dynamics for Deformable Object Manipulation, 2022.

[2] Gautam Salhotra, I-Chun Arthur Liu, Marcus Dominguez-Kuhne, and Gaurav S. Sukhatme. Learning Deformable Object Manipulation from Expert Demonstrations, 2022.


**Quality Of The Limitations Section:**

Limitations are addressed clearly

**Reviewer Expertise:**

4: The reviewer is confident but not absolutely certain that the evaluation is correct

**Robotics Focus:**

Sufficient demonstration on hardware

**Strengths And Weaknesses:**

**Strengths:**
* The method can be easily implemented and used for any Learning from Demonstration (LfD) methods.
* The approach is demonstrated on a robotic system.
* Figures are clear and useful for understanding the method and results.

**Weaknesses:**
* The main problem with this paper is the lack of comparison between the proposed method and state-of-the-art point-cloud-based or non-point-cloud-based methods. Moreover, most baselines are variants of your method (Direct Vector, Dense Transformation) or generic methods (Image CNN), and they are too weak to be considered competitive methods. A competitive point-cloud-based method could be [1]. Some competitive non-point-cloud-based methods include SAC, SAC-DrQ, SAC-CURL from Softgym, or a recent LfD method implemented in Softgym [2].
* The evaluation metric (success rate / success rate of the demonstrator) used in the experiments may not be suitable. It is possible for reinforcement learning (RL) and BC+RL methods to exceed the demonstrator’s performance. Even though the experiments do not contain these methods (which I think they should), it is better to show the unnormalized success rate in experiments.
* Behavior Cloning (BC) is known to suffer from the distributional shift problem. Since BC is utilized in this framework, how do you deal with this problem?


**Summary Of Recommendation:**

I recommend weak reject in light of the "strengths and weaknesses" discussion above. The main issue of this paper is that it is not clear how this method compares to state-of-the-art point-cloud-based methods and non-point-cloud-based methods. Due to the lack of this comparison, it is not possible to draw the conclusion that learning from point clouds using ToolFlowNet is superior to learning from RGB or depth images.

---

> ### Author Response · Authors · 2022-08-27
> **Response to Reviewer LESm (Part 1 of 2)**
>
> Dear Reviewer,
>
> We thank you for your time and for the paper feedback. We have revised the paper draft based on your suggestions with changes highlighted, and have included new experiments. We have attached the revised PDF to this post. Below, we respond to your questions:
>
> > **Q1**: “The main problem with this paper is the lack of comparison between the proposed method and state-of-the-art point-cloud-based or non-point-cloud-based methods. Moreover, most baselines are variants of your method (Direct Vector, Dense Transformation) or generic methods (Image CNN), and they are too weak to be considered competitive methods. A competitive point-cloud-based method could be [1]. Some competitive non-point-cloud-based methods include SAC, SAC-DrQ, SAC-CURL from Softgym, or a recent LfD method implemented in Softgym [2].”
>
> **A1**: Thank you for the helpful suggestions on alternative methods. To clarify, in this paper we study **imitation learning** (IL) using Behavioral Cloning **from point clouds**, with different action representations. The focus of the work is thus distinct from reinforcement learning (RL) approaches which study a different problem setting. As mentioned in the introduction, learning from point clouds is interesting in its own right due to its preservation of 3D data and potentially lower Sim2Real gap.  Point cloud based methods can be robust to variations in lighting, colors, textures, and shadows, which both RGB and RGBD-based methods are susceptible to; further, unlike monocular image-based methods, point-cloud based methods can distinguish between a large object that is far from the camera and a small object that is close to the camera.
>
> In any case, to get a rough sense of how RL compares against IL, as suggested, we have performed experiments with **SAC-CURL** from SoftAgent on the **PourWater** and **ScoopBall** environments using RGB image inputs. The results can be found in **the supplement, Section B.5**, and also on the project website with quantitative and qualitative RL results in the section titled **"[Baseline: Reinforcement Learning (SAC-CURL)](https://sites.google.com/view/point-cloud-policy/home#h.ino1wnq46ps8)"**. We show these results for both the original environments as well as the newly added 6 DoF action spaces.
>
> To summarize, our results show that even after **1 million environment interaction steps**, SAC-CURL gets **0 or near 0 raw success rate** on PourWater (for both 3 DoF and 6 DoF action spaces), whereas IL is significantly more reliable with raw success rates of 0.720 and 0.544 for the two action spaces, respectively. On the task of ScoopBall, for the 4 DoF action space, RL achieves 0.891 success, outperforming ToolFlowNet’s 0.728 success. Nonetheless, SAC-CURL requires 400K environment interaction steps to exceed the performance of IL (see Figure S2, left subplot). In the 6 DoF action space, ToolFlowNet’s performance of 0.952 exceeds that of SAC-CURL (0.788) after 1 million steps. Furthermore, as shown on the website in the “[Baseline: Reinforcement Learning (SAC-CURL)](https://sites.google.com/view/point-cloud-policy/home#h.ino1wnq46ps8)” section, the policies from RL produce jerky and discontinuous behavior as compared to policies from IL. The IL policies also do not require online environment interaction and only use a small set of offline demonstrations. We use just 25 demonstrations for 6 DoF ScoopBall (due to the different ladle making the task easier), and 100 demonstrations for 4 DoF ScoopBall, and 100 demonstrations for both action spaces for PourWater. Still, we emphasize that our main contribution is to study **imitation learning** (IL) using Behavioral Cloning **from point clouds**, with different action representations.
>
> We agree that the suggested references [1] and [2] are relevant. We cited ACID [1] in the Related Works; however, ACID studies using a dynamics model to manipulate a 3D deformable object to a provided target shape. Their goal-conditioned manipulation problem setting is fundamentally distinct from ours as we are focused on action representations for (non-goal conditioned) Behavioral Cloning, and it is unclear how we can make a proper comparison between ACID and ToolFlowNet.
>
> The paper by Salhotra et al. [2] is another impressive work, and we have added this as a related work citation at the end of Section 2. This method studies combining imitation learning and reinforcement learning, so it requires online environment interaction and, furthermore, uses RGB image input. Thus, it also studies a different problem setting as we do since our main focus is imitation learning from point clouds; thus we do not consider it crucial to directly compare to this method.

---

> ### Author Response · Authors · 2022-08-27
> **Response to Reviewer LESm (Part 2 of 2)**
>
> > **Q2**: The evaluation metric (success rate / success rate of the demonstrator) used in the experiments may not be suitable. It is possible for reinforcement learning (RL) and BC+RL methods to exceed the demonstrator’s performance. Even though the experiments do not contain these methods (which I think they should), it is better to show the unnormalized success rate in experiments.
>
> **A2**: First, we would like to clarify that we allow the normalized success rate to exceed 1 if the method exceeds the demonstrator’s performance; this occurs for our method (ToolFlowNet) for the ScoopBall 4D task, in which we report a normalized success rate of 1.152.  Still, we agree that it is useful to also report un-normalized success rates. We have added the un-normalized success rates in Section B.4, **Table S6**.
>
> > **Q3**: Behavior Cloning (BC) is known to suffer from the distributional shift problem. Since BC is utilized in this framework, how do you deal with this problem?
>
> **A3**: We agree that distributional shift is an important problem as you point out. The main purpose of this paper is to study Behavioral Cloning methods from point cloud input and demonstrate the benefit of tool flow as an action representation. To address distributional shift, a method like DAgger could help. ToolFlowNet is compatible with approaches like DAgger; employing DAgger or a similar technique is an orthogonal direction to our work.
>
> > **Q4**: Why does “ToolFlowNet, No Skip Conn.” have a success of 0% in the PourWater task? How would skip connections help when learning to predict rotations? The authors should explain this more clearly (lines 243 - 245).
>
> **A4**: We thank the reviewer for this interesting question. If there are no skip connections, then in the upsampling procedure of segmentation PointNet++ (i.e., the interpolation layers), the same latent vector (obtained by global pooling) is copied to every point, so the final predicted flow is the same for every point. When SVD converts such a flow to a transformation, this results in a translation-only transformation with no rotation. We have added a clarification of this detail to the supplementary material, Section B.4.
>
> > **Q5**: Consider re-writing the first sentence (lines 17 to 19). The first sentence of the introduction should be clear and concise.
>
> **A5**: Thank you for the suggestion. We have rewritten this sentence and split it into two.
>
> > **Q6**: [3.2] It’s not clear why the ground truth transformation is not used in the consistency loss. Consider adding this information on line 154.
>
> **A6**: We have added a clarification after this line. To elaborate: the consistency loss is a function of two items: (1) a set of points and (2) a set of flow vectors on those points. From the flow vectors, it deduces a transformation via SVD and then applies that transformation on the set of points. Then it compares the resulting induced flow vectors from this application of the transformation with the original flow vectors before SVD. Thus, the ground truth action is never used in the consistency loss.
>
> > **Q7**: It is a good idea to show the success rate of the demonstration data in Table 1.
>
> **A7**: We agree that showing the demonstrator success rate is a good idea. We have put the demonstrator success rate for the four combinations of environments and action spaces in Tables S6 and S7 in the supplement. The reason is that these tables report the raw success rate and thus it is likely easier to match raw numbers from the policy performance with the raw demonstrator success rate. We have put the demonstrator success rate after the “Demo:” text in the four relevant columns. In contrast, Table 1 of the main text shows the normalized success rate, in which the demonstrator would have a normalized success rate of 1.
>
> _We greatly appreciate your feedback, and please let us know if these changes have addressed your concerns and if there are other questions you may have._

---

### Author Response · Authors · 2022-08-28
**Unified Response to Reviewers: Revised Paper + Appendix**

Dear Reviewers,

_We sincerely thank you for the helpful feedback_. We have responded to all reviewers below and incorporated edits to the main text and to the supplementary material. The changes are noted in highlighted text for ease of convenience. Please see attached the revised PDF to this post. Note that this PDF reports a new set of experiments (in supplement, Section B.6) compared to the PDFs initially posted in responses to Reviewers LESm, ckzp, and Vzsj.

Once again, we deeply appreciate all your feedback, which we believe has improved the quality of the paper. We hope you enjoy consulting the revised version. Thank you for your time.

---

### Author Response · Authors · 2022-09-01
**Brief Follow-Up: Slight Addition to Supplement B.6**

Dear Reviewers,

We have added a new experiment to the paper, in Supplement B.6 with a **State (Learned) Direct Vector** baseline. This is a new baseline which still uses ground-truth state information (as in State (G.T.) Direct Vector), but instead of using it directly as input to an MLP, the state information is the target for a PointNet++ which takes segmented point clouds as inputs and then predicts the state information. This is done in a first round of training. Then this network (which we fix) predicts state information which is passed to an MLP in a second round of training to predict the actions. In sum, results for this new baseline are worse compared to State (G.T.) Direct Vector and ToolFlowNet. Supplement B.6 has further details.

Thank you for your time and attention.

---

### Meta-Review · Area_Chair_VnnE · 2022-08-10

**Recommendation:** Accept (Poster)
**Confidence:** 3

**Metareview:**

This paper presents an imitation learning approach for manipulation using tools based on the tool’s point cloud flow. The authors claim that using point cloud results in better performance than using transformation. The method is demonstrated with 2 tasks x 2 action spaces in simulation. One of the tasks is also implemented on hardware. Overall, the proposed method performs best on average, if not the best in every scenario.

The authors addressed most of the reviewer comments, including new examples with 6D action space. If the paper is accepted, the authors should clarify the role of tool pose in the training phase (cf. A2 and A3 in the response to reviewer sFXa). Also, a deeper discussion on why using point cloud is better than using transformation is appreciated given that prior works have successfully used quaternions and vector-angle representation.

**Best Paper Nomination:**

No

---

> ### Author Response · Authors · 2022-08-28
> **Response to Meta-Reviewer**
>
> Dear Meta-Reviewer,
>
> We thank you and the reviewers for all the helpful comments.
>
> We have responded to all reviewers and incorporated edits to the main text and to the supplementary material. The changes are noted in **highlighted text** for ease of convenience. Please see attached the revised PDF.
>
> Below we respond to three reviewer concerns:
>
> > **Q1**: Lack of comparison to SOTA RL methods (primarily from Reviewer LESm)
>
> **A1**: We agree that comparison to alternative SOTA methods is important. However, the focus of our paper is on **action representations for behavioral cloning from point clouds**, which is somewhat orthogonal to the comparison of imitation learning vs reinforcement learning. Nonetheless, we ran additional reinforcement learning experiments and reported them in the revised draft, in Section B.5 of the supplement, showing that RL methods either fail (pouring tasks) or require at least 40X more data just to reach the performance of ToolFlowNet (scooping tasks).
> We have elaborated further in the response to Reviewer LESm.
>
> > **Q2**: Limited evaluation: Reviewers sFXa and Vzsj requested more complex tasks with 6D action spaces.
>
> **A2**: We have performed a new set of experiments based on the existing environments (scooping and pouring) that use 6D action spaces. The results can be found in Table 1 and demonstrate that ToolFlowNet remains the most effective method on average.
>
> > **Q3**: Need one-to-one mapping of the points in point clouds? (from Reviewer sFXa)
>
> **A3**: Thank you for raising this point. To clarify, estimating or requiring a one-to-one mapping across time is **not necessary and none of our experiments in the paper** used an estimated one-to-one mapping (correspondence) in point clouds at different time steps. We have elaborated further in our response to Reviewer sFXa.
>
> We have replied to reviewers in more detail to individual points. Please let us know if you have any additional concerns or feedback. We sincerely appreciate the advice that you and the reviewers have given to help make our paper better.